# Augmented Efficacy of Uttroside B over Sorafenib in a Murine Model of Human Hepatocellular Carcinoma

**DOI:** 10.3390/ph15050636

**Published:** 2022-05-22

**Authors:** Mundanattu Swetha, Chenicheri K. Keerthana, Tennyson P. Rayginia, Lekshmi R. Nath, Nair Hariprasad Haritha, Anwar Shabna, Kalishwaralal Kalimuthu, Arun K. Thangarasu, Sreekumar U. Aiswarya, Somaraj Jannet, Sreekumar Pillai, Kuzhuvelil B. Harikumar, Sankar Sundaram, Nikhil Ponnoor Anto, Dee H. Wu, Ravi S. Lankalapalli, Rheal Towner, Noah Isakov, Sathyaseelan S. Deepa, Ruby John Anto

**Affiliations:** 1Division of Cancer Research, Rajiv Gandhi Centre for Biotechnology, Thiruvananthapuram 695014, Kerala, India; swetham@rgcb.res.in (M.S.); keerthanack@rgcb.res.in (C.K.K.); rayginia@rgcb.res.in (T.P.R.); lekshmirnath@aims.amrita.edu (L.R.N.); harithahnair@rgcb.res.in (N.H.H.); shabnaa@rgcb.res.in (A.S.); kalimuthu@rgcb.res.in (K.K.); aiswaryaus920@gmail.com (S.U.A.); jannets@rgcb.res.in (S.J.); harikumar@rgcb.res.in (K.B.H.); 2Department of Biotechnology, University of Kerala, Thiruvananthapuram 695011, Kerala, India; 3Department of Pharmacognosy, Amrita School of Pharmacy, Amrita Vishwa Vidyapeetham, AIMS Health Science Campus, Kochi 682041, Kerala, India; 4Chemical Sciences and Technology Division, CSIR-National Institute for Interdisciplinary Science and Technology, Thiruvananthapuram 695019, Kerala, India; arunkumarchem43@gmail.com (A.K.T.); ravishankar@niist.res.in (R.S.L.); 5Department of Surgical Oncology, Jubilee Mission Medical College and Research Institute, Thrissur 680005, Kerala, India; sreekumarpillai16959@gmail.com; 6Department of Pathology, Government Medical College, Kottayam 686008, Kerala, India; sankarradhika@rediffmail.com; 7The Shraga Segal Department of Microbiology, Immunology and Genetics, Faculty of Health Sciences, Ben-Gurion University of the Negev, P.O. Box 653, Beer Sheva 84105, Israel; antop@post.bgu.ac.il (N.P.A.); noah@bgu.ac.il (N.I.); 8Section of Medical Physics, Department of Radiological Sciences, University of Oklahoma Health Sciences Center, Oklahoma City, OK 73104, USA; dee-wu@ouhsc.edu; 9School of Computer Science, Gallogly College of Engineering, University of Oklahoma, Norman, OK 731019, USA; 10School of Electrical and Computer Engineering, Gallogly College of Engineering, University of Oklahoma, Norman, OK 731019, USA; 11Departments of Pathology and Pharmaceutical Sciences, University of Oklahoma Health Sciences Center, Oklahoma City, OK 73104, USA; rheal-towner@omrf.org; 12Department of Biochemistry and Molecular Biology, and Stephenson Cancer Center, University of Oklahoma Health Sciences Center, Oklahoma City, OK 73104, USA; deepa-sathyaseelan@ouhsc.edu

**Keywords:** uttroside B, sorafenib, hepatocellular carcinoma, chemotherapeutic, apoptosis

## Abstract

We previously reported the remarkable potency of uttroside B (Utt-B), saponin-isolated and characterized in our lab from *Solanum nigrum* Linn, against HCC. Recently, the U.S. FDA approved Utt-B as an ‘orphan drug’ against HCC. The current study validates the superior anti-HCC efficacy of Utt-B over sorafenib, the first-line treatment option against HCC. The therapeutic efficacies of Utt-B vs. sorafenib against HCC were compared in vitro, using various liver cancer cell lines and in vivo, utilizing NOD.CB17-Prkdcscid/J mice bearing human HCC xenografts. Our data indicate that Utt-B holds an augmented anti-HCC efficacy over sorafenib. Our previous report demonstrated the pharmacological safety of Utt-B in Chang Liver, the normal immortalized hepatocytes, and in the acute and chronic toxicity murine models even at elevated Utt-B concentrations. Here, we show that higher concentrations of sorafenib induce severe toxicity, in Chang Liver, as well as in acute and chronic in vivo models, indicating that, apart from the superior therapeutic benefit over sorafenib, Utt-B is a pharmacologically safer molecule, and the drug-induced undesirable effects can, thus, be substantially alleviated in the context of HCC chemotherapy. Clinical studies in HCC patients utilizing Utt-B, is a contiguous key step to promote this drug to the clinic.

## 1. Introduction

According to the GLOBOCAN database, liver cancer is the third leading cause of cancer death worldwide in 2020, with approximately 830,000 deaths [1]. Sorafenib, the FDA-approved drug for HCC, is an orally administered multi-kinase inhibitor that inhibits cell proliferation via the inhibition of the serine/threonine kinase, RAF [2,3,4]. Moreover, sorafenib targets pro-angiogenic VEGF and PDGFR [5,6]. Sorafenib is widely studied and numerous experimental data, including those from clinical trials, reveal the appropriate reason for this drug to be administered as a first-line option to treat HCC patients. However, drug-induced side effects and poor patient survival hampers the efficacy of sorafenib chemotherapy [7,8]. Our discovery of Utt-B, a saponin derived from the plant black nightshade (*Solanum nigrum* Linn), as a candidate drug against HCC, has attracted global recognition by gaining multi-national patents [USA (US2019/0160088A1), Canada (3,026,426), Japan (JP2019520425), Korea (KR1020190008323)], and commercial technology transfer (Q Biomed, Inc., New York, NY, USA). Our previous studies have revealed a complexity in Utt-B pharmacodynamics where we report that the drug induces an apoptotic mode of cell death in HCC cells but simultaneously functions as an inducer of autophagic signals, the inhibition of which further enhances the therapeutic index of Utt-B against HCC [9]. Importantly, Utt-B has now gained the ‘orphan drug’ status as designated by the U.S. FDA.

So far, no reports have been disseminated on the biological activity of Utt-B apart from our studies. However, there are notable efforts on the optimization on Utt-B extraction from plants using sophisticated chromatography and spectrometry techniques [10,11,12]. This demands the necessity for further studies to advance this drug to the clinics as a treatment option for HCC in clinical patients. The current study intends to compare the in vitro and in vivo chemotherapeutic efficacies of Utt-B with that of sorafenib. Our data highlight the therapeutic supremacy of Utt-B over sorafenib in a murine model of human HCC.

## 2. Results

We previously demonstrated the potency of Utt-B to drive liver cancer cells to death under in vitro conditions [13]. An evaluation of sorafenib under identical conditions revealed the comparative efficacies of the two drugs in the induction of liver cancer cytotoxicity. Amongst the various liver cancer cell lines that were employed for the screening, HepG2 was the most sensitive to the two drugs in a cell viability assay and comparison of the drugs’ IC50 (0.5 µM and 5.8 µM, respectively) indicates that Utt-B is approximately 10-fold more potent than sorafenib (Figure 1A–C). Our previous report revealed that Utt-B is not cytotoxic to the normal immortalized hepatocytes, Chang Liver, even at high drug doses [13]. In contrast, treatment using high concentrations of sorafenib resulted in the death of Chang Liver, indicating that Utt-B is safer than sorafenib to normal hepatocytes (Figure 1D). Nuclear condensation of HepG2 cells upon drug treatment indicated that Utt-B is a stronger apoptosis-inducer, when compared to sorafenib (Figure 1E). HepG2-induced colony formation was drastically inhibited by Utt-B compared to sorafenib, which had a reduced effect, indicating the superior anti-clonogenic potential of Utt-B (Figure 1F). Furthermore, flow cytometry analysis revealed that Utt-B drove a larger number of cells to apoptosis in comparison to sorafenib (Figure 1G). A similar observation was made in the wound healing assay where Utt-B-treated cells exhibited a reduced wound closure compared to sorafenib-treated cells, suggesting that Utt-B possesses a better anti-migratory potential over sorafenib (Figure 1H). We also tested the effect of Utt-B on the activation of the initiator and executioner caspases −9 and 7, along with the induction of PARP cleavage, which are indicative of triggered apoptotic machinery [14]. In concordance with our previous observation, Utt-B induced the efficient cleavage of procaspases and PARP [13]. Notably, Utt-B potentiated enhanced cleavage of these apoptosis markers than sorafenib (Figure 1I–K).

A critical xenograft study involving Utt-B and sorafenib revealed their efficacies in vivo. The schematics of the human HCC-induced tumor development and the drug treatment regimen in the NOD-SCID murine models have been provided in Figure 1L and detailed in the methodology. Following the drug treatment, the tumours were procured for further analysis. Firstly, the tumour volume in the Utt-B- and sorafenib-treated mice was drastically reduced in comparison to that of the control mice. However, a striking difference in the tumour size was noted, when comparing the Utt-B- and the sorafenib-treated mice, suggesting that Utt-B displays an upper hand in the destruction of HCC cells (Figure 1M,N).

Utt-B treatment accentuated the cleavage of executioner caspase 7, in vivo (Figure 2A). Histopathological analysis of the excised tumour samples from the control group revealed marked mitosis of tumour cells, while Utt-B treatment induced massive apoptosis, authenticating the above observations (Figure 2B). Furthermore, TUNEL staining revealed the presence of a higher number of apoptotic cells in the Utt-B-treated mice in comparison to the sorafenib-treated mice (Figure 2C). The immunohistochemical analysis in the tumor tissues for the expression status of cell proliferation markers, PCNA and Ki67, and the apoptosis marker, cleaved PARP, revealed the augmented potency of Utt-B over sorafenib (Figure 2D).

We previously reported a detailed toxicological evaluation of Utt-B in vivo, demonstrating its pharmacological safety in Swiss albino mice [13]. Though a well-studied chemotherapeutic drug, we conducted a toxicity evaluation of sorafenib under identical conditions, since some mice in the sorafenib-treated group of the xenograft study exhibited toxicity symptoms such as paralysis of the legs and faecal blood stains towards the end of the treatment period. A schematic of the treatment regimen for the toxicity analysis of sorafenib and Utt-B in Swiss albino mice is depicted in Figure 2E. However, the analysis of acute and sub-chronic liver toxicity studies revealed that the indicated dosages of sorafenib neither inflicted any behavioral changes nor any shifts in the hematological parameters, renal profile, or liver function, except that there is a significant difference in ALP levels in the acute toxicity analysis of 5xIC50 doses of sorafenib (175 mg/kg), in comparison to the untreated control mice, attesting the biological safety of sorafenib in normal Swiss albino mice, at the IC50 concentration (Figure 2F–I). Hence, the previously observed toxicological symptoms in the xenograft study may be attributed to the immune deficiency of NOD-SCID mice. Similar observations have been reported in CB17/16 SCID mice [15]. We also noted severe breathing problems, signs of hypertrophy, and binucleation, representing regenerative changes in liver hepatocytes in Swiss albino mice treated with 5xIC50 doses of sorafenib (Figure 2J). On the contrary, only reversible micro-vesicular fatty changes associated with chemotherapy were observed in liver tissues of the mice treated with even up to five times the IC50 dose of Utt-B, which exactly correlates with our previous report [13]. Together, our data demonstrate the superior therapeutic efficacy that Utt-B holds over sorafenib, in the attenuation of HCC.

## 3. Discussion

Our previous study screened cancer cells of various origins for their cytotoxicity towards Utt-B, wherein, the HCC cells exhibited remarkable cytotoxic potential [13]. Our very recent study has unraveled, with mechanism-based evidence, better modes to further improve the chemotherapeutic potential of Utt-B against HCC [9]. Except for a couple of in silico observations, there are no reports on the biological activity of Utt-B, other than ours, to date. A molecular docking study involving Utt-B demonstrated the ability of the drug to bind Human coronin-IA protein, which plays a critical role in regulating actin dynamics and cargo internalization [16]. In addition, the in silico study from another group has shown that Utt-B can be docked to human vimentin, a recognized marker for epithelial–mesenchymal transition, well-known for its role in the maintenance of cellular integrity and stress resistance [17]. However, biological validation of these in silico observations has not been conducted. The current study depicts Utt-B as a better anti-HCC chemotherapeutic in comparison with sorafenib, the standard drug used to treat the disease. Our previous studies demonstrated the exceptional anti-HCC efficacy of Utt-B, which is a strong inducer of caspase-mediated apoptosis [9,13]. However, a comparative evaluation of the anti-HCC potency of Utt-B with sorafenib, the most studied US FDA-approved drug against HCC, was a requisite for better apprehension [18]. Sorafenib is known to induce side effects that often result in poor patient survival or recurrence of HCC [19,20,21,22]. Though our evaluation of sorafenib-associated toxicological parameters in normal Swiss albino mice revealed that the drug is safe at a dose corresponding to its in vitro IC50 concentration, higher dosages of sorafenib resulted in severe toxicity in mice as per the acute and sub-chronic toxicity studies. Moreover, the immunocompromised mice that were used for the induction of human HCC xenografts in the present study also exhibited symptoms of toxicity, even at the dose corresponding to the IC50 of sorafenib. Several other studies have also reported sorafenib-associated chemoresistance [23,24]. On the contrary, we have already shown that Utt-B is safe to mice even at elevated drug concentrations, authenticating the pharmacological safety of Utt-B over sorafenib [13]. Since Utt-B has been granted the status of an ‘orphan drug’ against HCC by the US FDA, the current study was critical to ensure a comparative screening of this drug with a prominent FDA-approved HCC drug, sorafenib. For instance, a recent study utilized a transcriptomics-based drug repurposing approach to comparatively evaluate the growth inhibitory effects of novel drugs against sorafenib-resistant HCC [25]. The human liver cancer cell line, HepG2 was used as a tool for the in vivo HCC model, due to its enhanced sensitivity towards both drugs, as per our previous observations [9,13]. Our data depict that Utt-B possesses credibility to function as a better anti-HCC drug than sorafenib. Currently, in vivo studies comparing the efficacy of Utt-B and sorafenib are being pursued in murine orthotopic models of human HCC.

## 4. Materials and Methods

### 4.1. Chemicals

Isolation and purification of Utt-B were done as previously reported [13]. Important cell culture reagents such as Dulbecco’s Modified Eagle Medium (DMEM) (GIBCO, 12800-017), Minimum Essential Medium (MEM) (61100061), Foetal Bovine Serum (10270-106), and streptomycin sulfate (GIBCO, 11860-038) were obtained from Invitrogen Corporation (Grand Island, NE, USA). Poly Excel HRP/DAB detection system universal kit (PathnSitu Biotechnologies Pvt. Ltd., Secunderabad, India, OSH001) was used for immunohistochemistry experiments. MTT reagent was purchased from TCI Chemicals (India) Pvt. Ltd. (Chennai, India) (D0801). DAPI (D9542), Propidium Iodide (P 4170), RNase A (10109142001), antibodies against β-actin (12620S), GAPDH (8884S), Caspase 9 (9508S), Caspase 7 (12827S) PARP (9532S), and cleaved PARP (5625S) were obtained from Cell Signaling Technologies (Beverly, MA, USA). Annexin V apoptosis detection kit (sc4252AK) and the antibodies against PCNA (sc25280) and Ki67 (sc23900) were purchased from Santa Cruz Biotechnology (Santa Cruz, CA, USA). DeadEnd™ Colorimetric TUNEL System was procured from Promega (G7132). MycoSensor qPCR Assay Kit used to test mycoplasma contamination in cells was purchased from Agilent technologies. All other chemicals were purchased from Sigma Chemicals (St. Louis, MO, USA), unless otherwise mentioned. 

### 4.2. Cell Culture

The liver cancer cell lines, HepG2 and Huh-7 were purchased from ATCC and Hep3B, SK-HEP-1, and immortalized normal Chang Liver cells were from NCCS, Pune. MTT assay was performed in cancer lines as previously described. Chang Liver cells were cultured in Minimum Essential Medium (MEM) supplemented with 10% Fetal Bovine serum. All other cell lines HepG2, Huh-7, Hep3B, and SK-HEP-1 were cultured in Dulbecco’s Modified Eagle Medium (DMEM) supplemented with 10% Foetal Bovine serum. All the cell lines used in this study were mycoplasma-tested and were free from mycoplasma contamination.

### 4.3. In Vitro Assays

Immunoblotting, clonogenic, and wound healing assays were performed as previously described [9]. Immunoblotting analysis was performed as described previously [13] and the quantification of the blots were carried out using ImageJ software. Immunofluorescence microscopy for the detection of nuclear condensation and apoptosis, using DAPI and Annexin V, respectively, were performed as previously described [9].

### 4.4. Animals

In vivo experiments were conducted according to the RGCB animal ethical committee approval (IAEC/634/RUBY/2017, IAEC/588/RUBY/2017, and IAEC/810/RUBY/2020). Mice were housed in a 12 h light/dark cycle with access to food and water. Tumor xenografts were generated by the subcutaneous injection of 5 × 10^6^ HepG2 cells in matrigel into the right lower flank of 6-week-old NOD-SCID (NOD.CB17-Prkdcscid/J) male mice. Two weeks post-injection, the mice were separated into 3 groups (*n* = 6). Group I received the vehicle, Group II and Group III received an intraperitoneal injection of Utt-B in PBS (10 mg/kg body weight) and sorafenib in cremophor vehicle (35 mg/kg body weight), respectively, on alternate days for 1 month, followed by the euthanasia of animals and subsequent procurement of the tissue samples for further analyses. Histology, immunohistochemistry, and TUNEL assays were performed in the tumor samples as previously described [9]. The in vivo toxicity analysis (acute and sub-chronic) of sorafenib and Utt-B was conducted in 6–8-week-old Swiss albino mice (male and female, 1:1). The test group was administered with varying concentrations of sorafenib (0, 35, and 175 mg/kg body weight) and Utt-B (0, 10, and 50 mg/kg). After the culmination of the study period, the liver function and renal profile of the mice were analysed. 

### 4.5. Flow Cytometry

Annexin-PI staining was performed to get the percentage of apoptotic cells as previously reported using BD FACS Diva software Version 5.0.2 (BD Biosciences, 2350 Qume Dr, San Jose, CA 95131-1807, USA) [9].

### 4.6. Statistical Analysis

Data analysis was performed using GraphPad Prism software Version 8 (GraphPad Software Inc., San Diego, CA, USA). Student’s *t*-tests, One-way ANOVAs and Two-way ANOVAs were used to evaluate the statistical significance. *p <* 0.05 was considered as statistically significant. The error bars represent ± SD and are indicative of three independent experiments.

## 5. Conclusions

The present study highlights the therapeutic supremacy of Utt-B over sorafenib, the widely administered FDA-approved anti-HCC drug. We demonstrate the superior anti-clonogenic potential and anti-proliferating efficacy of Utt-B against liver cancer cells, compared to sorafenib in vitro. The pro-apoptotic potential of Utt-B is evident from the enhanced cleavage of caspases and significant increase in the percentage of apoptotic cells in Utt-B-treated HCC cells in comparison with sorafenib. Xenograft studies in immunocompromised murine models of human HCC exhibited a significant reduction in tumor development, with negligible side effects, in Utt-B-treated animals, whereas sorafenib-treated mice exhibited symptoms of severe toxicity.

Undesirable effects associated with sorafenib chemotherapy is an impending problem in HCC patients. The present study attests to our previous report that Utt-B is pharmacologically safe up to five times the IC50 dose in acute and sub-chronic toxicity models, while even the IC50 dose of sorafenib is toxic to immunocompromised mice, and elevated doses of sorafenib induce adverse effects such as breathing problems, signs of hypertrophy, and mild regenerative changes in liver hepatocytes, in normal immune competent mice. Together, we highlight Utt-B as a promising anti-HCC drug, owing to its enhanced therapeutic efficacy and pharmacological safety over sorafenib, the first-line treatment option for HCC.

## 6. Patents

Title of the invention: Uttroside B and Derivatives Thereof as Therapeutics for Hepatocellular Carcinoma. The invention has been granted patent by the USA (US2019/0160088A1), Canada (3026426.), Japan (JP2019520425), and Korea (KR1020190008323).

## Figures and Tables

**Figure 1 pharmaceuticals-15-00636-f001:**
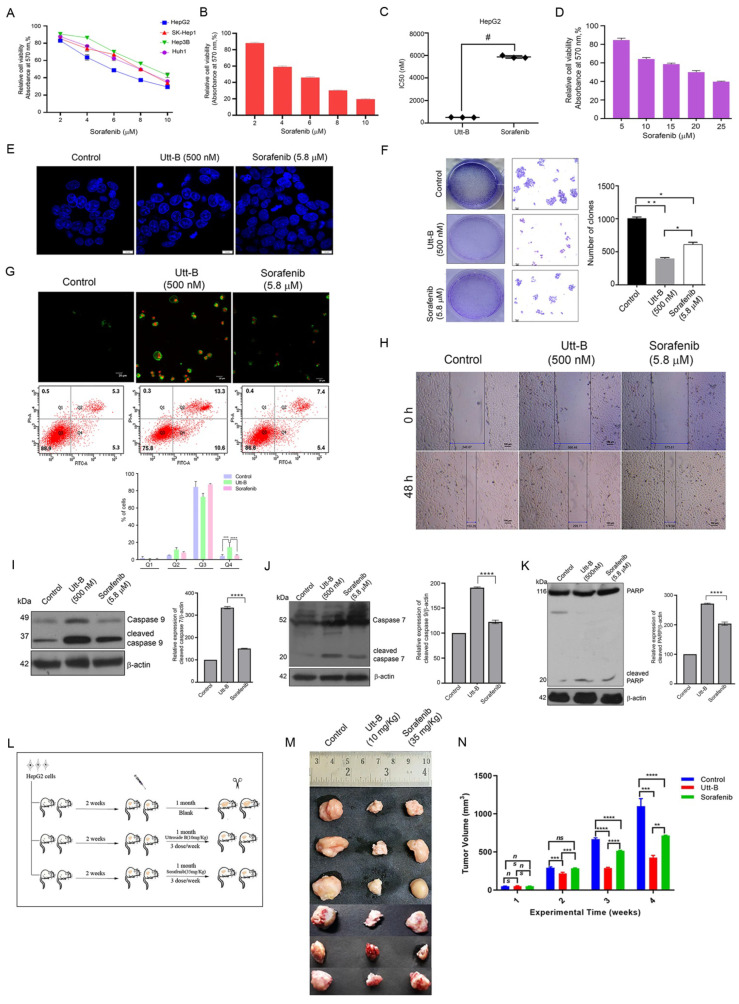
Utt-B triggers apoptotic mode of cell death in HCC, exhibiting better therapeutic efficacy compared to sorafenib (**A**) MTT assay of sorafenib in the indicated liver cancer cell lines. (**B**) Assessment of sorafenib cytotoxicity in HepG2 cells, as assessed by MTT Assay. (**C**) A plot between Utt-B vs. sorafenib treatment, in vitro, # *p* ≤ 0.001, significant. (**D**) Cytotoxicity analysis of sorafenib in normal hepatocytes, Chang Liver. (**E**) DAPI staining indicating that Utt-B induces more nuclear condensation in HepG2 cells, than sorafenib. (**F**) Clonogenic assay reveals an augmented anti-clonogenic potential of Utt-B than that of sorafenib. One-way ANOVA was used for statistical analysis, * *p* ≤ 0.1; ** *p* ≤ 0.01. (**G**) Annexin-PI flow cytometric analysis shows an increase in apoptosis of HCC cells upon Utt-B treatment. Two-way ANOVA was performed for statistical analysis. (**H**) Wound healing assay showing the augmented anti-migratory potential of Utt-B than that of sorafenib. (**I**–**K**) Immunoblot analysis demonstrates an enhancement in cleavage of caspase 9, 7, and PARP in HepG2 cells treated with Utt-B, in comparison to sorafenib. Statistical analysis was done using One-way ANOVA, **** *p* ≤ 0.0001. (**L**) A scheme depicting the induction of human HCC tumors in NOD-SCID mice using HepG2 cells. (**M**) An image of xenograft-excised tumors post-drug treatment regimen. (**N**) A graphical representation of tumor volumes of indicated treatment groups. Two-way ANOVA was used for statistical analysis, *** *p* ≤ 0.001, ** *p* ≤ 0.01.

**Figure 2 pharmaceuticals-15-00636-f002:**
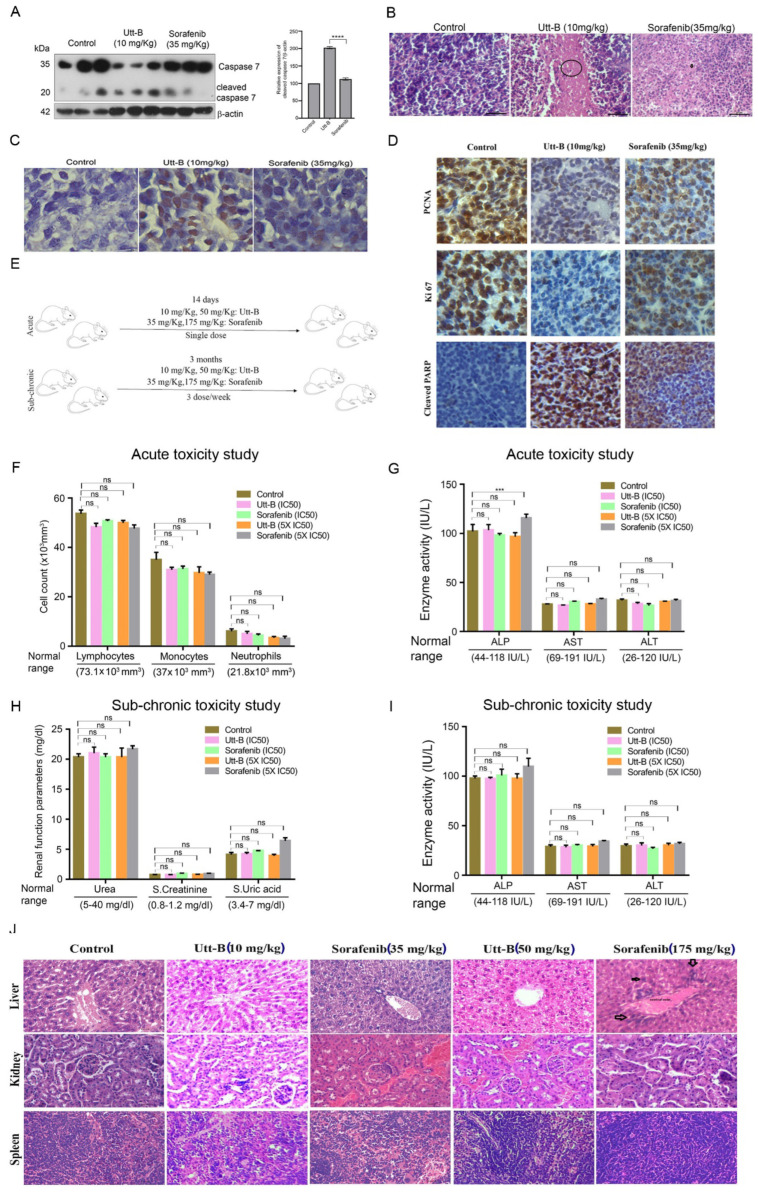
Augmented anti-HCC efficacy and pharmacological safety of Utt-B over sorafenib, in vivo. (**A**) Immunoblot analysis in tumor samples demonstrates an enhanced cleavage of caspase 7 upon Utt-B treatment, in comparison to sorafenib-treated. One-way ANOVA was performed for statistical analysis, **** *p* ≤ 0.0001. (**B**) Histopathological staining of xenograft-derived tumors from treatment groups. (**C**) Increase in TUNEL-positive cells following Utt-B treatment confirms the augmentation of apoptosis. (**D**) Immunohistochemical analysis of nuclear proliferation markers PCNA, Ki67, and the apoptosis marker, cleaved PARP in tumor sections. (**E**) A scheme of toxicity analysis of sorafenib and Utt-B, in Swiss albino mice. Statistical analysis was done using Two-way ANOVA. (**F**–**I**) Acute and sub-chronic toxicity analysis of sorafenib and Utt-B treated mice groups (Control, IC50 and 5XIC50), *** *p* ≤ 0.001, ns non significant. (**J**) Histopathological analysis of liver, kidney, and spleen tissues of mice post-sorafenib and post-Utt-B regimen.

## Data Availability

Raw data compliant with the institutional confidentiality policies are available upon request. Data requests should be sent to the corresponding author. Data is contained within the article.

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
