# Peer review of "Augmented Efficacy of Uttroside B over Sorafenib in a Murine Model of Human Hepatocellular Carcinoma"

_pharmaceuticals, 2022, doi:10.3390/ph15050636_

Round 1
Reviewer 1 Report
The manuscript studied the efficacy of uttroside B compared to sorafenib using in vivo and invitro models
- Introduction and discussion are not informative. Was uttroside B tested in other cancer models? was the model used for HCC similar to that of Human? Has uttroside B been reported for any activity other than cancer? What is the role of caspase 7/ caspase 9 in cancer?
- Figure 1 H, statistical analysis for flow cytometry results should be added
- Figure 2 B, J, changes observed should be indicated on figures as arrows, arrow head, etc
- What was the age of animals used?
- Graphical abstract is recommended
Author Response
Dear Reviewer,
Thank you for reviewing our manuscript “Augmented efficacy of uttroside B over sorafenib in a murine model of human hepatocellular carcinoma” for publication in Pharmaceuticals, section Pharmacology (Manuscript ID: 1663646). We are happy to answer your comments and concerns. We have modified the manuscript as per your suggestions and the corresponding changes have been represented as track changes in the revised manuscript. We hope that the revised manuscript is acceptable for publication in Pharmaceuticals.
To address your suggestion to incorporate quantification of flow cytometry analysis, we have made significant changes in the panel arrangement of Figure 1. Figure 1 E, which was previously representing Clonogenic assay has now become Figure 1 F. Figure 1 F, which was previously representing nuclear condensation study, has now changed to Figure 1 E. Figure 1 G, which was previously representing wound-healing assay has now become Figure 1 H. Figure 1 H, which was previously representing Flow cytometry analysis has now become Figure 1 G. The graph representing quantification of flow cytometry analysis is included in Figure 1 G. All these changes are given as track changes in the revised manuscript.
Thank you for your careful effort to consider our work.
Comments and Suggestions for Authors
The manuscript studied the efficacy of uttroside B compared to sorafenib using in vivo and in vitro models
Comment 1: Introduction and discussion are not informative.
Response: Thank you for the suggestion.
We have modified the introduction and discussion as per the reviewer’s suggestion in the revised manuscript.
Comment 2: Was uttroside B tested in other cancer models?
Response: Thank you for the query.
Uttroside B was screened for its cytotoxicity against a panel of seven human cancer cells of different origin viz. skin cancer (A375), liver cancer (HepG2), colon cancer (HCT-116), leukemia (HL60), cervical cancer (HeLa), breast cancer (MDA-MB-231), and lung cancer (A549). Since HepG2 (liver cancer) cells showed maximum sensitivity to this compound with an IC50 of 500 nM, we selected liver cancer cells for further studies. This has been reported in our previous paper Nath et al., 2016 Sci Rep. We have mentioned this in the revised manuscript as per the reviewer’s suggestion (Discussion, Page 10).
Comment 3: Was the model used for HCC similar to that of Human?
Response: We implemented the human hepatocellular carcinoma cell line, HepG2 for the in vitro and in vivo setting owing to its superior sensitivity to the drugs, Utt-B and sorafenib in comparison to all other human HCC cell lines investigated. Currently, we are standardizing Orthotopic murine models of human HCC. We have mentioned this in the revised manuscript as per the reviewer’s suggestion (Discussion, Page 10).
Comment 4: Has uttroside B been reported for any activity other than cancer?
Response: Till now there are no wet-lab results on any activity of Utt-B, other than our report on its anti-HCC efficacy. Two molecular docking studies have revealed that Utt-B binds to Human coronin-IA, which plays critical roles in regulating actin filament dynamics and cargo internalization and vimentin, which is known to maintain cellular integrity and provide resistance against stress and is recognized as a marker for epithelial-mesenchymal transition. We have incorporated this information in the revised manuscript (Discussion-Page 10). Moreover, our lab is now exploring its efficacy as an immunomodulator and as a potent drug against non-alcoholic steatohepatitis (unpublished data).
Comment 5: What is the role of caspase 7/ caspase 9 in cancer?
Response: Thank you for the comment.
As per our previous study, Utt-B activates both initiator and effector caspases (Nath et al., 2016). In the present study we verified the results by picking a candidate from each class of caspases and compared the effect of Utt-B with sorafenib. We have mentioned this in the revised manuscript as per the reviewer’s suggestion (Results, Page 4).
Comment 6: Figure 1 H, statistical analysis for flow cytometry results should be added
Response: Thank you for the suggestion.
A graph depicting the quantification of flow cytometry analysis has been included in figure 1G of the revised manuscript. Two-way ANOVA was used for analysis. (Results - page 7)
Comment 7: Figure 2 B, J, changes observed should be indicated on figures as arrows, arrow head, etc
Response: Thank you for the suggestion.
We have marked the Histopathology images with circle, arrow and arrow head.
Comment 8: What was the age of animals used?
Response: Thank you for the comment. We have already mentioned this in the ‘Materials and Methods’, Section-Animals, Line number ( ) on Page 3. For anti-tumor study, 4- 6 week old NOD-SCID (NOD.CB17-Prkdcscid/J) male mice were used. For Acute and sub-chronic toxicity studies 6-8 week old Swiss Albino mice of both sexes were used. We have marked the portion in the revised manuscript as per the reviewer’s suggestion.
Comment 9: Graphical abstract is recommended
Response: Thank you for the suggestion.
Graphical abstract is included in the revised manuscript.
Thanking you
Yours sincerely,
Ruby John Anto.

Reviewer 2 Report
The manuscript entitled "Augmented efficacy of uttroside B over sorafenib in a murine model of human hepatocellular carcinoma" presents in vitro and in vivo data, that (along with previous publications of the authors) support the use of uttroside B as alternative therapy against hepatocellular carcinoma (HCC). The data regarding the superior cytotoxicity and induction of apoptosis on HCC of uttroside B over sorafenib are solid and convincing, especially considering that uttroside B is achieving these effects although it is used in lower concentrations than sorafenib.
However, the data regarding the pharmacological safety are incomplete, since the authors in this manuscript evaluate the safety of sorafenib only. Then, they compare these observations with other ones from their previous publications (Nath et al., 2016), although they were performed in different setting. Hence, the conclusion that Utt-B is a promising anti-HCC drug, "owing to its enhanced therapeutic efficacy and pharmacological safety over sorafenib" (quoting lines 240-241 of the manuscript) is not acceptable based on these data. The authors have to repeat the experiments presented in Fig. 2E, F-I, and J evaluating in parallel the selected doses of sorafenib and uttroside B. This is the only acceptable methodology to support the safety of a treatment over another.
Author Response
Dear Reviewer,
Thank you for reviewing our manuscript “Augmented efficacy of uttroside B over sorafenib in a murine model of human hepatocellular carcinoma” for publication in Pharmaceuticals, section Pharmacology (Manuscript ID: 1663646). We are happy to answer the comments made by you, and we have made the corresponding changes as track changes in revised manuscript. We hope that this revised manuscript is now acceptable for publication in Pharmaceuticals.
To address the suggestion of Reviewer 1 to incorporate quantification of flow cytometry analysis, we have made significant changes in the panel arrangement of Figure 1. Figure 1 E, which was previously representing Clonogenic assay has now become Figure 1 F. Figure 1 F, which was previously representing nuclear condensation study, has now changed to Figure 1 E. Figure 1 G, which was previously representing wound-healing assay has now become Figure 1 H. Figure 1 H, which was previously representing Flow cytometry analysis has now become Figure 1 G. The graph representing quantification of flow cytometry analysis is included in Figure 1 G. All these changes are given as track changes in the revised manuscript.
Thank you for your careful effort to consider our work.
Comments and Suggestions for Authors
The manuscript entitled "Augmented efficacy of uttroside B over sorafenib in a murine model of human hepatocellular carcinoma" presents in vitro and in vivo data, that (along with previous publications of the authors) support the use of uttroside B as alternative therapy against hepatocellular carcinoma (HCC). The data regarding the superior cytotoxicity and induction of apoptosis on HCC of uttroside B over sorafenib are solid and convincing, especially considering that uttroside B is achieving these effects although it is used in lower concentrations than sorafenib.
However, the data regarding the pharmacological safety are incomplete, since the authors in this manuscript evaluate the safety of sorafenib only. Then, they compare these observations with other ones from their previous publications (Nath et al., 2016), although they were performed in different setting. Hence, the conclusion that Utt-B is a promising anti-HCC drug,"owing to its enhanced therapeutic efficacy and pharmacological safety over sorafenib" (quoting lines 240-241 of the manuscript) is not acceptable based on these data.
Comment 1: The authors have to repeat the experiments presented in Fig. 2E, F-I, and J evaluating in parallel the selected doses of sorafenib and uttroside B. This is the only acceptable methodology to support the safety of a treatment over another.
Response: Thank you for the suggestion. We have already compared the pharmacological safety of Utt-B and sorafenib (both acute & sub-chronic) as per the protocol number IAEC/588/RUBY/2017. The results obtained for Utt-B were identical to that of the previous report (Nath et al., 2016). Since the present manuscript is submitted as a ‘Brief Report’, in Pharmaceuticals, which allows only two figures for this format, we omitted the data with Utt-B, due to the space constraints. As per the reviewer’s suggestion, we have incorporated the data and have modified Figure 2 to illustrate the comparison of Utt-B and sorafenib (Fig. 2 E-J) in the revised manuscript.
Thanking you
Yours sincerely,
Ruby John Anto.
Reviewer 3 Report
The manuscript entitled “Augmented efficacy of uttroside B (Utt-B) over sorafenib in a murine model of human hepatocellular carcinoma” and authored by Swetha et al reported the therapeutic supremacy of Utt-B over sorafenib, the widely-administered FDA-approved anti-HCC drug. Authors demonstrated the superior anti-clonogenic potential and anti-proliferating efficacy of Utt-B against liver cancer cells, compared to sorafenib in vitro. In xenograft studies in immunocompromised murine model of human HCC, authors reported dramatic reduction in tumor development, with negligible side effects, in Utt-B-treated animals, whereas sorafenib–treated mice exhibited symptoms of severe toxicity. Before focusing on uttroside B, a saponin isolated from Solanum nigrum, general health-promoting benefits of natural products should be sufficiently introduced. If considered, the following studies could fulfill such a requirement: PMID: 33338743, PMID: 32837538, PMID: 33782460, PMID: 32460808. It would be useful if patents and clinical trials that relate to sorafenib combined with biomolecules are discussed. Other than that, and after full revision, the manuscript could be of interest for journal’s readers.
Other comments
1. What is the source of the NOD.CB17-Prkdcscid/J male mice used in this study? If purchased, more details are needed then.
2. In section “4.4 Animals”, quote “…followed by the euthanasia of animals and subsequent procurement of…”, how exactly were mice euthanized?
3. What was the basis of selection of the in vivo doses used for Utt-B and sorafenib? How pharmacologically and physiologically relevant such doses are?
4. In figure 1D, the cytotoxicity analysis of Utt-B in normal hepatocytes, Chang Liver should be plotted against that presented now for sorafenib. Although effects of examined drugs were reported for caspase 7 and 9, they were not assessed for caspase 3, why? In fact, sorafenib has been reported to affect caspase 3 both in HCC and other cancer types in vitro (PMID: 32610149) and in vivo (US Patent 10,568,873 & US Patent 10,933,076). In figure 1H, the flow cytometry data should be quantified. Figure 1J is not clear, another gel should be used. In addition, ALL uncropped gels should be provided in supplementary documents.
5. Figure 2A show great variation in the levels of cleaved caspase 7, a better representative gel should be used. Panels C and D should also be quantified.
6. Unlike what figure 2 title do state “Augmented anti-HCC efficacy and pharmacological safety of Utt-B over sorafenib, in vivo”, panels E-J focus only on sorafenib, NOT Utt-B. Parallel analyses should be conducted or the figure title must be modified to be more reflective of the figure’s presented data.
Author Response
Not applicable
Reviewer 4 Report
In this study, the authors found that uttroside B (Utt-B) holds an augmented anti-HCC efficacy than sorafenib. Higher concentrations of sorafenib induce severe toxicity in Chang Liver, as well as in acute and chronic in vivo models.
Comments:
- What's the target and mechanism of Utt-B anti-HCC?
- More HCC cell lines not only HepG2 should be used to compare the anti-HCC effects of Utt-B and sorafenib.
- The results of Figure 1F-H and 2 C-D should be quantified.
- In the Figure 1K, the molecular weight of cleaved PARP detected by it antibody (CST 5625S) should be 89KDa.
- Why were the backgrounds of tumors different in the Figure 1M? Were these tumors gotten and taken picture at the different times?
Author Response
Not applicable
Round 2
Reviewer 1 Report
The authors addressed my commments
Reviewer 2 Report
The revised version of the manuscript is suitable for publication.
Reviewer 3 Report
The revised manuscript is much improved.
Reviewer 4 Report
No